# Creation of a prototype biomimetic fish to better understand impact trauma caused by hydropower turbine blade strikes

Ryan Saylor[1,2], Peter L. Wang[3], Mark Bevelhimer[1], Peter Lloyd[3], Jesse Goodwin[3], Robert Laughter[3], David Young[3], Dustin Sterling[1], Paritosh Mhatre[3], Celeste Atkins[3] and Brian Post[3]

[1] Environmental Science Division, Oak Ridge National Laboratory, Oak Ridge, TN, United States of America
[2] Bredesen Center for Interdisciplinary Research and Graduate Education, University of Tennessee, Knoxville, TN, United States of America
[3] Manufacturing Demonstration Facility, Oak Ridge National Laboratory, Oak Ridge, TN, United States of America

Corresponding author
Ryan Saylor, saylorr@ornl.gov

## ABSTRACT

Biomimetic model organisms could be useful surrogates for live animals in many applications if the models have sufficient biofidelity. One such application is for use in field and laboratory tests of fish mortality associated with passage through hydropower turbines. Laboratory trials suggest that blade strikes are especially injurious and often causes mortality when fish are struck by thinner blades moving at higher velocities. Dose-response relationships have been created from these data, but the exact relationship between fish mortality and the actual forces enacted on fish during simulated blade strike testing remains unknown. Here, we describe the methods used to create a prototype biomimetic model fish composed of ballistic gelatin and covered with a surrogate skin to better approximate the biomechanical properties of a fish body. Frozen fish were scanned with high-fidelity laser scanners, and a 3D-printed, reusable mold was created from which to cast our gelatin model. Computed tomography scan data, imaged directly or taken from online data repositories, were also successfully used to create CAD models for use in additive manufacturing of molds. One 3-axis accelerometer was embedded into the gelatin to compare accelerometer data to dose-response data from previous laboratory research on live fish. The resulting model (*i.e.*, Gelfish) had a statistically indistinguishable tissue durometer to that of real fish tissue and preliminary blade strike impact testing suggested its overall flexibility was similar to that of live fish. Gelfish was designed with biofidelity as its guiding principle and our results suggest initial experimentation was successful. Future research will include replication of initial Gelfish test results, quantitative measurement of model flexibility relative to real fish, and inclusion of surrogate skeletal structures to enhance biofidelity. Use of more sophisticated sensors would also better quantify the physical forces of blade strike impact and help determine how said forces correlate with rates of mortality observed during tests on live fish.

## INTRODUCTION

The field of biomimetics often produces revolutionary inventions and innovations that overcome persistent engineering challenges—many such breakthroughs are the result of studying aquatic organisms. Fishes and marine mammals are at the center of many studies because of their unique adaptations to sense the environment and move efficiently to obtain resources while evading potential threats (*Fish, 2006*; *Triantafyllou, Weymouth & Miao, 2016*; *Salazar, Fuentes & Abdelkefi, 2018*). For example, the lateral line of fish and sensory structures of marine mammals have inspired instrumentation that allows for enhanced navigation in water (*Yang et al., 2010*; *Kottapalli et al., 2016*). Similarly, fish shape and body movement have been replicated to better overcome hydrodynamic drag, which allow ships and submersible vehicles to move more efficiently in the water (*Fish & Kocak, 2011*). Detailed study of the integumental complex and fins have also led to innovations that enhance movement efficiency and maneuverability of marine vessels (*Lauder et al., 2016*; *Wainwright & Lauder, 2017*). Many of these innovations are used to design aquatic robots with onboard sensors, fin-like structures, and body shapes that provide more efficient movement through the water (*Zhang et al., 2007*; *Serchi, Arienti & Laschi, 2013*; *Salazar, Fuentes & Abdelkefi, 2018*; *Hosseini, Tabrizi & Meghdari, 2019*). Fish scales are also well-studied because the imbrication patterns and material properties provide flexibility and puncture resistance against predatory attacks, which has been applied to body armor development (*Browning, Ortiz & Boyce, 2013*; *Yang et al., 2013*; *Zhu et al., 2013*; *Sherman et al., 2017*). Production of biomimetics continues and has helped overcome many engineering obstacles, but innovations are also needed to help protect the organisms that serve as inspiration.

Laboratory research using live organisms is obligatory in certain studies, but biomimetic models may serve as a useful substitute for live organisms in others. Vertebrate animals specifically are used in a multitude of laboratory and field studies (*Bennett et al., 2016*; *Cooke et al., 2016*; *Couto & Cates, 2019*; *Sloman et al., 2019*), but fishes form the basis of much laboratory research (*Borski & Hodson, 2003*; *Lawrence et al., 2009*). Legal use of fishes in federally funded scientific research must meet rigorous animal welfare standards monitored by the Institutional Animal Care and Use Committee (IACUC) in the USA (*Lawrence et al., 2009*; *Bennett et al., 2016*). In addition to the cost of acquiring or producing live animal subjects, research budgets must also include care and maintenance which is expensive and time consuming. High standards of animal care ensure regulatory compliance and is also essential for scientific purposes because laboratory animals must be healthy, accurate representatives of the population. Studies involving fish often require dozens if not hundreds of individuals to properly account for natural variability, which further increases the financial burden of animal husbandry. In certain circumstances, the desired fish species may not be available because it is rare, difficult to capture or keep alive in captivity, or protected by state and federal laws. To successfully receive authorization to use live animals, researchers are usually required to explain why use of an animal model (or computer simulation) is not possible and many fields are beginning to substitute animal models where possible (*Sloman et al., 2019*). In most cases, it is difficult to mimic

or recreate an organism without studying it first, but the loop could be closed by creating a biomimetic model for future use in place of live animals. Combined, these facts suggest that if a biomimetic model existed, with sufficiently high biofidelity, the need for live animals would be less necessary in certain fields.

One application for a biomimetic model fish would include field and laboratory tests related to concerns of fish passage through hydropower turbines. There are nearly 2500 hydroelectric dams in the USA (*EIA, 2020*) and many riverine fishes are at a particularly high risk of turbine passage due to their migratory behavior (*Pracheil et al., 2016b*; *Silva et al., 2018*). Live fish are often used during passage survival testing that is a part of the relicensing process for conventional hydropower dams. Hydropower facilities must undergo relicensing every 30–50 years and hundreds of dams are projected to submit relicensing applications within the next decade (*Uria-Martinez et al., 2021*). Exact passage conditions of fish are generally unknown and have relied on insights from computational fluid dynamic (CFD) models to estimate probability of exposure to turbine passage stressors. Impacts from turbine blade runners are one of the most injurious stressors and laboratory tests on live fish suggest it may cause organ damage, skeletal fractures, amputation, and death (*Bevelhimer et al., 2019*; *Saylor, Fortner & Bevelhimer, 2019*; *Saylor et al., 2020*). Rates of injury and death are highest with thinner blades, higher impact velocities, and when struck on the lateral surface near the center of gravity of a fish (*Turnpenny et al., 1992*; *EPRI , 2008*; *Bevelhimer et al., 2019*; *Saylor, Fortner & Bevelhimer, 2019*; *Amaral et al., 2020*). Dose–response relationships generated from these laboratory trials are an important resource for designing more fish-friendly turbines; however, these data are limited in scope to just a few fish species exposed to what is presumed to be the worst-case impact scenarios. Furthermore, technology like the hard-bodied autonomous Sensor Fish, that records actual hydraulic conditions from within a functioning turbine (*Carlson, Duncan & Gilbride, 2003*; *Deng et al., 2007a*; *Deng et al., 2007b*; *Deng et al., 2014*), is available but incapable of sufficiently mimicking responses of live fish impacted by turbine blades. To that end, a biomimetic fish would be a useful surrogate for live animal tests because it could be used more than once and be validated using previously generated dose–response data.

Herein we detail the methods used to create a prototype biomimetic model fish (hereafter referred to as Gelfish) composed of ballistic gelatin and containing an embedded sensor. We used 3D scanning and imaging technologies to successfully replicate the general shape and surface features of multiple fish species. Scanned images were used to additively manufacture a reusable mold from which to cast the ballistic gelatin model. Ballistic gelatin was chosen for our initial model because of its extensive use as a tissue simulant in ballistic testing. Tissue durometer (firmness) of the Gelfish was compared to real fish tissue to assess biofidelity of our model. Durometer was chosen because it is easy to measure and is well-established in medicine to assess changes in tissue (*Falanga & Bucalo, 1993*; *Cuaderes et al., 2009*; *Moon et al., 2012*) and organ (*Belyaev et al., 2010*) hardness caused by disease, or to confirm biofidelity of cosmetic surgery (*Brown, Brown & Murphy, 2017*; *Murphy et al., 2020*), which suggests it is a viable option for fish tissue as well. In addition, preliminary observations of model flexibility were also compared to live fish to better assess Gelfish

biofidelity following a simulated turbine blade strike. To our knowledge, 3D printing molds instead of the animal model directly, has not been applied to the production of a whole-organism biomimetic model before. More specifically, the objectives of this study were to (1) test the ability of ballistic gelatin to match whole-body firmness of fish tissues, (2) quantify how preparation temperature and warming time affect gelatin durometer, (3) determine efficacy of Plasti Dip® as a surrogate fish skin, (4) additively manufacture molds and cast gelatin models for at least five species of fish, (5) embed a 3-axis accelerometer into Gelfish to record characteristics of blade strike impact, and (6) compare Gelfish responses to available data from live fish when exposed to simulated blade strike impacts to help assess model biofidelity.

## MATERIALS & METHODS

### Ballistic gelatin experiments

To our knowledge, there are no published accounts of ballistic gelatin being used as a surrogate for fish tissue, so we designed several experiments to establish its baseline material properties. Ballistic gelatin was chosen because of its established use as a human and animal tissue simulant in ballistics research (*Jussila, 2004*; *Maiden et al., 2015*). Furthermore, there are well established protocols and recipes for ballistic gelatin that were easy to modify to meet our needs. We used ballistic gelatin powder specifically formulated to simulate human body density (Vyse® Professional Grade Ballistic Gelatin; Lot #12953; Custom Collagen, Inc., Addison, Illinois, USA; customcollagen.com) for all trials and final model preparation. Our main metric to measure the material properties of gelatin was tissue durometer (*i.e.,* material hardness or resistance to indentation) for all ballistic gelatin trials. More specifically, we measured durometer with a Shore Type-OO durometer (Model DD-4 Digital Durometer; Precision = ± 0.1 units; Rex Instruments, Buffalo Grove, Illinois, USA; durometer.com) which is best suited to measure soft gels and animal tissue. An automated stand (Model OS-1 Operating Stand, Rex Instruments, Buffalo Grove, Illinois, USA; durometer.com) lowered the meter to the sample at precisely the same rate under a consistent load pressure for all samples, thereby decreasing measurement error.

In preliminary trials, we tested two methods of ballistic gelatin preparation that were modified from other sources to accommodate our smaller sample volumes (*Jussila, 2004*; *Cronin, 2011*; *Maiden et al., 2015*). Method one (referred to as cooling hydration) included heating deionized water to a desired temperature using a water bath (Thermo Scientific Precision Microprocessor Controlled 280 Series Water Bath; thermofisher.com), followed by adding the heated water into a large (~900 mL) polypropylene container containing gelatin powder. The water and gelatin were then mixed with a metal spatula until completely homogenized so that no clumps remained. At this point, up to 150 µL of de-foaming agent (Custom Collagen, Inc., Addison, Illinois, USA; customcollagen.com) was added to remove foam and excess bubbles. The mixture then cooled to room temperature (~22 °C) which allowed the gelatin to hydrate. After this cooling hydration period, the container was covered with a lid and refrigerated for 12 h at 4 °C to allow the gelatin to completely set. Finally, the block of ballistic gelatin could be removed, cut into pieces, re-melted, and

distributed as needed into other containers for testing. The second method (referred to as heated hydration) was similar to the previous except the heated water and gelatin mixture was covered with parafilm wax, placed back into the same temperature water-bath, and allowed to hydrate at this temperature for at least 10 min. Following the heated hydration period, the gelatin mixture could be distributed into test containers, allowed to cool to room temperature, and refrigerated for 12 h at 4 °C. We preferred the heated hydration method because it allowed the gelatin mixture to hydrate without cooling, avoided evaporative water loss during re-melting, and samples could be poured immediately into test containers. Both methods produced comparable estimates of durometer in our ballistic gelatin samples, but heated hydration was preferred because of more consistent heating and avoided unnecessary re-melting. Lastly, we used cinnamon oil to increase the shelf-life of our ballistic gelatin samples well beyond the normal 7 to 11-day limitation imposed on use of refrigeration alone (*Jussila, 2004*; *Staymates & Gillen, 2010*). We used cinnamon oil (NOW® Cinnamon Cassia oil; Item #051210; gnc.com) dissolved in 95% ethanol (1:10) at a concentration of 515 ppm as a microbial growth inhibitor. Cinnamon oil was dissolved in 95% ethanol to make it more miscible in water because pure cinnamon oil extract will separate from the gelatin (*Jussila, 2004*). Use of the heated hydration method and cinnamon oil ensured more consistent durometer measurements during experimental trials.

Most published accounts of ballistic gelatin include use of 10 or 20% solutions (mass to volume) of gelatin powder dissolved in deionized water (*Jussila, 2004*; *Cronin & Falzon, 2011*); however, we were unsure which concentration best mimicked fish tissue. We tested a total of five concentrations including 10, 15, 20, 25, and 30% to determine which concentration best approximated the durometer of actual fish tissue (see last experiment). Each ballistic gelatin concentration was prepared in triplicate. A heated hydration protocol with a preparation temperature of 65 °C was used to create each gelatin mixture. Following hydration, ~60 mL of each replicate was added to a 100-mL, polystyrene weigh boat and allowed to cool at room temperature. When the gelatin reached room temperature (denoted by solidification of the gelatin) all samples were labelled, placed into a large, 33× 38 cm, 6-Mil plastic storage bag, and refrigerated overnight. Following refrigeration, three randomly selected sample weigh boats were removed and allowed to warm to room temperature for 30 min. Ten durometer measurements were recorded for each sample by removing it from the weigh boat, flipping it over, and taking measurements across the gelatin's bottom surface. The durometer measurements for each replicate were averaged and the arithmetic mean of all three represented the average durometer of each concentration group.

There are conflicting accounts of which water temperature is best for preparation of ballistic gelatin with respect to maintaining optimal material properties of gelatin. Preparation temperatures may range from 40 to 90 °C or higher; however, manufacturers recommend temperatures near 40 °C to maintain its tissue-simulating properties (*Fackler & Malinowski, 1987*; *Cronin & Falzon, 2011*). We also experimented with preparation temperature to determine how it affected the durometer of our ballistic gelatin samples. All ballistic gelatin samples in these experiments were made using a 25% gelatin concentration. Three temperature treatments—45, 55, and 65 °C—were prepared in triplicate and

durometer was measured for each sample. In addition, we also tested how warming time following refrigeration affects durometer measurements. This experiment included preparation of three replicate gelatin samples using a concentration of 25% and a water temperature of 45 °C. Following refrigeration, five durometer measurements were made immediately (time 0), every 10 min up to 1 h, then 15 min up to 2 h, and finally every 30 min for up to 4 h. Durometer was measured and reported in the same manner as the concentration experiment for each temperature and warming time treatment group.

The final set of experiments compared the material properties of ballistic gelatin to that of an actual fish to determine how closely we could mimic natural tissue. In addition to gelatin, we tested the use of an artificial skin surrogate that covered our ballistic gelatin samples. More specifically, commercially available Plasti Dip® as a skin surrogate and found that spraying was preferable over dipping to sufficiently cover the gelatin samples. The first set of experiments was used to determine if the surrogate skin covering would significantly increase durometer compared to uncovered samples. We prepared an additional three replicates of 25% ballistic gelatin at 45 °C for use in these tests. After refrigeration, we allowed samples to warm for 30 min and took 10 durometer measurements. One layer of surrogate skin was then applied to the gelatin sample and allowed to cure for 30 min under a fume hood. The samples were then refrigerated for an additional 12 h, after which they were removed, allowed to warm for 30 min, and durometer measurements were taken. The same protocol was repeated for two, three, and four additional layers of surrogate skin for comparison. Next, we collected durometer data from three bluegill sunfish, *Lepomis macrochirus*, with a total length of ~16 cm and mass of ~90 g. All three fish were euthanized *via* overdose of 250 ppm clove oil in 95% ethanol (1:10) immediately prior to durometer measurements. Durometer was taken for each bluegill at 27 different locations along the entire body surface except the head which was mostly bone and the fins which were too thin to measure (Fig. 1). Another set of 27 durometer measurements were taken on the same three fish after removing scales from the entire lateral surface. The shapes of the Gelfish models and bluegill specimens required us to take durometer measurements by hand because both surfaces were curved which precluded use of the automated stand used for other ballistic gelatin experiments. In this way, we created data sets for bluegill whole-fish durometer with and without scales for comparison of ballistic gelatin with and without a skin surrogate. Average durometer was reported in the same manner as the concentration experiment for each surrogate skin layer sample, Gelfish model, and bluegill tested.

All statistical tests were performed using *R v.4.0.2* statistical programing language and Sigma Plot v12. One-way analyses of variance (ANOVA) were used to compare the average durometer of (1) different ballistic gelatin concentrations prepared at 45 °C and (2) preparation temperature groups composed of 25% ballistic gelatin. A one-way repeated measures ANOVA was used to analyze the difference in average durometer between warming time and skin-layer treatment groups. Paired t-tests were used to compare average durometer between (1) Gelfish with and without Plasti Dip skin and (2) bluegill sunfish with and without scales, whereas an unpaired *t*-test was used to compare average durometer between Gelfish with skin and bluegill sunfish with scales. In the event a significant difference was detected by ANOVA, we used Benjamini–Hochberg post-hoc

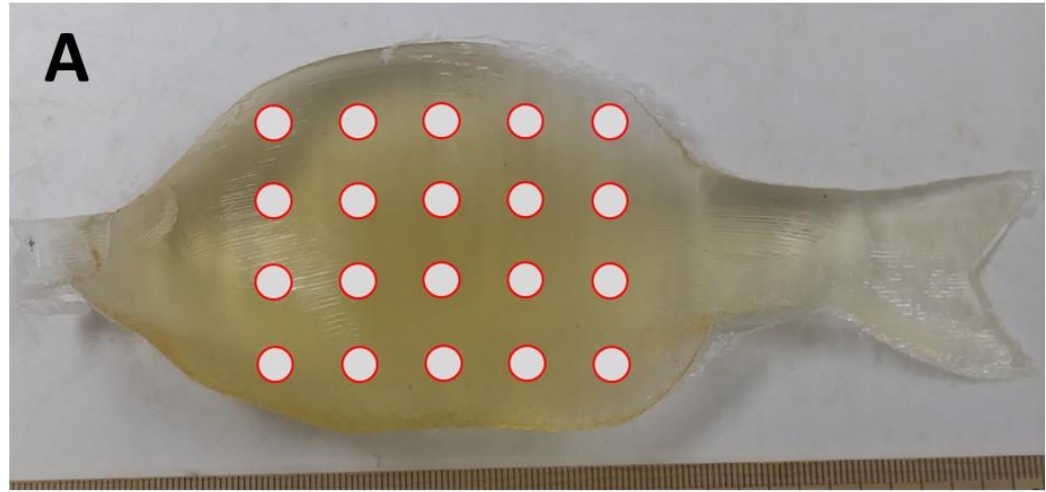

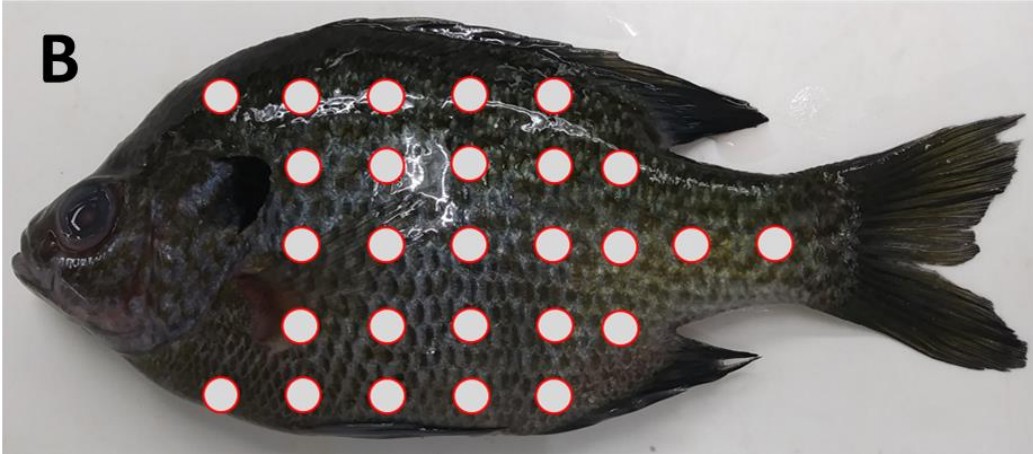

**Figure 1  Location of durometer measurements on Gelfish *versus* bluegill sunfish.**  Relative durometer measurement locations (circles) were taken on the left side of (A) Gelfish cast without skin and (B) bluegill with scales. Durometer measurements were replicated for both Gelfish and actual bluegill, *i.e.,* $n = 3$ for each. The Gelfish models and bluegill were 16 cm total length and 90 g mass. We also measured Gelfish with surrogate skin and bluegill without scales at the same approximate locations (not pictured).

multiple comparison tests to determine the statistical relationship between treatment groups. Finally, linear regression was used to test the relationship between ballistic gelatin concentration and average durometer. All statistical decisions were based on $\alpha = 0.05$.

## Fish scanning and image collection

Most biometric image data used to create our Gelfish model originated from 3D scans of freshly caught fish. We scanned four species of fish including bluegill sunfish, rainbow trout (*Oncorhynchus mykiss*), gizzard shad (*Dorosoma cepedianum*), and white bass (*Morone chrysops*) which varied in size (Table 1). These species were chosen because they represent the range of fish body shapes that may pass through hydropower turbines and because blade strike laboratory data are available for each species (*Pracheil et al., 2016a*; *Bevelhimer & Derolph, 2019*; *Bevelhimer et al., 2019*; *Saylor, Fortner & Bevelhimer, 2019*; *Saylor et al.,*

**Table 1 Size dimensions and scanning techniques used to create 3D images of each fish species.** The total length (TL) and wet mass (M) is included with each species scanned in this study.

| Common name | TL (cm) | M (g) | Scanning Method |
|---|---|---|---|
| Bluegill | 16.4 | 85.9 | FARO® SCANARM |
| Rainbow trout | 25.5 | 157.9 | Leica Laser Tracker & Scanner |
| Gizzard shad | 18.4 | 50.0 | Leica Laser Tracker & Scanner |
| White bass | 28.0 | 297.8 | FARO® SCANARM |
| American eel | 27.5 | 24.4 | Computed-tomography |

*2020*). To prepare for scanning, live fish were euthanized in an overdose of 250 ppm clove oil in 95% ethanol (1:10) for at least 15 min. Each fish was secured in an upright position with paired fins placed against the body and with the mouth and operculum closed. Individuals were frozen in this position at −20 °C for 12 h prior to scanning. Freezing was necessary to prevent movement of appendages during scanning which helped minimize image processing time. Additionally, the frozen fish was secured to a platform in an upright position that prevented movement but allowed for complete access to scan the entire fish. Finally, each fish was completely covered with a white, matte-finish spray paint to reduce surface reflections caused by fish scales. Two different scanners were used to capture fish images: a Leica Laser Tracker and Scanner (accuracy ± 0.060 mm; lieca-geosystems.com) and a FARO® SCANARM blue light laser scanner (accuracy ± 0.075 mm; faro.com). During scanning, Verisurf software (verisurf.com/software) was used to convert images into a point cloud file.

All laser-scanned point cloud data were processed and converted into a computer-aided design (CAD) model to be used for 3D printing. The point cloud data were converted to ASCII files and imported into Geomagic Design X (3dsystems.com) software. Some noise and unneeded areas (*i.e.,* scanning platform or fish restraint device) of the point cloud file were manually removed. The dorsal, pelvic, and anal fins were removed from the model to simplify preparation of the mold. We used internal software features like "reduce noise" with a smoothing level of 1 and default levels of "sampling" to smooth the point cloud data. After smoothing, we used the "wrap" command to transform the point cloud into a mesh. If the mesh contained more holes the images went through additional smoothing using the "fill holes" or "repair" features to close minor or larger gaps in the mesh, respectively. The final mesh was created by using "remesh", "smooth", and "remove spikes" features. Lastly, the final mesh was converted into a SolidWorks surface image, using the "auto surface" feature with the specifications of an organic geometry type, target patch count of 500, and default adaptive tolerance. The SolidWorks surface model was exported as a .STL file to be used in 3D printing of the fish mold.

We also investigated two additional forms of image acquisition including use of computed tomography scans of preserved specimens or from online databases. Our fifth and final species, American eel, *Anguilla rostrata*, was created by scanning a preserved specimen. The eel we used was much smaller than most sizes known to pass through hydropower turbines (Table 1); however, it was used to test our ability to account for and change fish size during image processing. The eel was scanned using a computed

tomography scanner through the Diagnostic Imaging Service available at the University of Tennessee College of Veterinary Medicine (UTCVM). The computed tomography scan of the eel was saved as digital imaging and communications in medicine (DICOM) file. An online digital repository called Morphospace (morphospace.org) was also used to find additional X-ray, computed tomography (also computed axial tomography; CAT), or laser-scanned images for white bass. While we generated our own 3D scan data, we were also interested in how readily available online data might also be used to create fish molds. Computed tomography images (either directly imaged or taken from repositories) are not available in point cloud form, so these images were first converted into .STL files using open source InVesalius 3 (invesalius.github.io) software. A contrast range with a lower bound between 42 and 65 and an upper bound of 255 best captured skin traits and underlying skeletal structures while simultaneously filtering out noise. Next, CloudCompare (daniel.gm.net/cc/) and MeshLab (meshlab.net) or Geomagic Design X were used to convert the .STL file into a point cloud by sampling one million points, which ensured sufficient detail for the CAD model while minimizing computational resources. The conversion of point cloud to a surface model (*i.e.*, an .STL file) followed the same procedure as that described above for laser-scanned images.

## Mold printing and construction

Each CAD model was further reviewed, and final modifications were made to ensure clean demolding and purging of air during casting. The thickness of the caudal fin and peduncle was increased so that the final cast model made of ballistic gelatin would not rip when removed from the mold. The bluegill and gizzard shad CAD models only included the caudal fin, whereas the rainbow trout and white bass CAD models also included slightly raised areas on the dorsal and ventral surfaces to represent dorsal and anal fins, respectively. Additional features found in all the species CAD models were raised areas that represented the eyes and operculum on the head which were also important landmarks for positioning sensors. The eel CAD model also went through additional processing to remove its notably longer dorsal, anal, and caudal fins. Other modifications to the eel model included scaling-up body proportions to account for different sizes of eel because the original fish was smaller than most eels known to pass through turbines. The fish CAD model was then subtracted from a box to create a negative space within it, which serves as the mold for the casting. A fill hole was added to each mold CAD model on the anterior (head region) through the mouth to avoid disrupting the shape of the body and allow easy access for filling the mold with ballistic gelatin. The final mold CAD model was split in half and alignment holes, pry points, and mounting hardware were added that ensured the mold was properly sealed and aligned during casting. To additively manufacture the molds, a build file was created that contains the printer tool path and material extrusion rates. We used the Stratasys Insight software to slice the molds into layers and generate these build files, which were loaded into the Stratasys Control Center for printing. Finally, molds were printed using a Stratasys Fortus 400mc printing system and were composed of spares infill acrylonitrile butadiene styrene (ABS). The inside of each half of the final

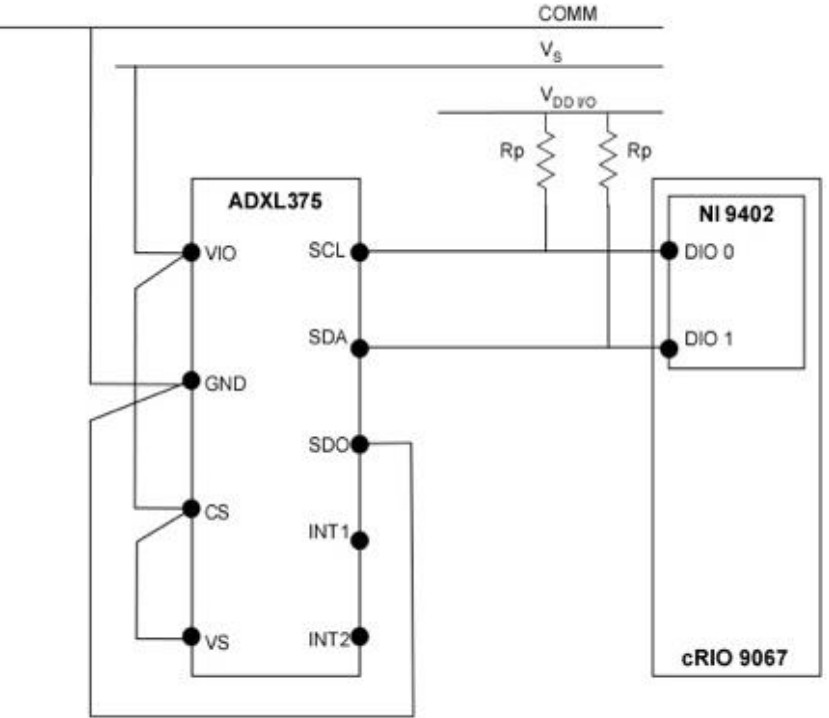

**Figure 2 Wiring schematic of the sensor and data acquisition system used in the Gelfish model.**
Wiring schematic of the single 3-axis accelerometer (ADXL375), data acquisition (cRIO 9067), and
interface system (NI 9402) used in the Gelfish model.

molds were polished with acetone to completely seal each surface prior to casting (*Sikder et al., 2014*; *Lalehpour & Barari, 2016*).

## Sensor specification and calibration

A 3-axis digital accelerometer (ADXL375, analog.com) with a capable measurement range of ±200 g was used for data acquisition in the ballistic gelatin model during simulated blade strike impact trials. Output data was accessed through the I²C interface at the rate of 800 Hz. The I²C protocol was configured with supply voltage $V_S = 3.3$V, interface voltage range $V_{DD/IO} = 3.3$V and external pull-up resistors $R_P = 1020\ \Omega$ (Fig. 2). The maximum pull-up resistor value ($R_{Pmax}$) was limited to 1180 Ω by the rise time ($t_r$) for SCL and SDA and the capacitive load on each bus line ($C_b$), which is given by the following equation:

$$R_{Pmax} = \frac{t_r}{0.8437 \times C_b} \tag{1}$$

Data acquisition consisted of NI cRIO 9067 (sine.ni.com), utilized as a target device and NI 9402 (sine.ni.com) module to provide the digital lines for SDA and SCL wires of I²C protocol. All hardware was programmed in LabVIEW (sine.ni.com) using the SPI and I²C Driver API, which served as the I²C master, and used the NI 9402 digital I/O to interface with the accelerometer. The LabVIEW Host code included in the API, in addition to the FPGA code, was used for initializing the accelerometer, configuring the I²C protocol

parameters, data read/write, and data logging operations. Data logging frequency was set *via* a timed loop in the host code and stored in a .TDMS file with a local time stamp associated with each reading. Calibration of the sensor was achieved by following the single point calibration scheme specified by the original equipment manufacturer. The 0g measurements represent a potential bias in acceleration that can result in incorrect output from the sensor, so 0g measurements were specified for all three axes. This calibration scheme aligned the *x*- and *y*- axes to the 0g field, while the *z*-axis was oriented to the 1g field. Alignment with the 1g field also required additional sensitivity compensation of the *z*-axis to ensure 0g was registered correctly. All 0g offset values were then stored in the LabVIEW code and written to the dedicated offset registers during sensor initialization. The wired accelerometer was potted with black epoxy potting compound (3M-DP270, 3m.com) using a custom mold. This provided the accelerometer, and the connections with the data acquisition system, necessary mechanical rigidity and watertight seal. The potted accelerometer was embedded into the ballistic gelatin model and could be used multiple times, *i.e.,* used in multiple ballistic models without deterioration.

## Gelfish model preparation & testing

For our initial complete Gelfish model, we chose to use rainbow trout because the body depth and width of this species could better accommodate an accelerometer. During Gelfish production, the accelerometer was held in place within the rainbow trout mold using a monofilament line that stretched from head to tail. We positioned the accelerometer posterior to where the operculum would be on a real fish. This location represents the mid-body area, which is associated with the highest rates of injury and mortality when fish are struck by hydropower turbine blades, including rainbow trout (*EPRI, 2008*; *Bevelhimer et al., 2019*; *Amaral et al., 2020*; *Saylor et al., 2020*). The mold was then securely closed and kept in an upright position to cast the mold. A 25% ballistic gelatin solution was prepared at 45 °C and injected into the mold using a 60-mL syringe with an extended tip. The syringe tip was inserted completely into the mold and gelatin was injected from the bottom upwards to avoid formation of bubbles. After it was filled, the ballistic gelatin was cooled for 10 min at room temperature, followed by refrigeration at 4 °C for 90 min. Refrigeration was used to accelerate cooling and decrease the time required for the gelatin to completely set. Following refrigeration, the ballistic gelatin model was removed from the mold, placed into a sealed plastic baggie, and refrigerated again at 4 °C overnight. A surrogate skin (*i.e.,* Plasti Dip®) was applied after overnight refrigeration such that four separate layers were added with at least 45 min of curing time between each layer. The final Gelfish model was then placed back into the baggie and refrigerated prior to its use in blade strike impact trials.

We used the same simulated blade strike apparatus and procedure described in (*Saylor et al., 2020*), to strike the rainbow trout Gelfish model. We struck the Gelfish 12 times, with each strike accounting for a different velocity and leading-edge blade width, as well as a different impact location and orientation on the model itself, while all blade strikes with the model occurred at 90° (Table 2). These strike conditions were chosen based on previous laboratory tests which found that mid-body, lateral strikes caused the highest rates

**Table 2 Blade strike impact conditions and changes in acceleration from trials performed on the rainbow trout Gelfish model.** Location of strike was mid-body (M) or tail (T) while orientation was lateral (L) or ventral (V). Alignment axis refers to which of three axes the Gelfish model aligned when held in place prior to blade strike testing. Impacts relative to the sensor were considered "Direct" if the blade contacted the model fish at the center of the accelerometer whereas "Indirect" strikes occurred when the blade made contact with the model posterior (towards the caudal fin) to the accelerometer.

| Trial No. | Blade width (mm) | Blade velocity (m/s) | Location (H, M, T) | Orientation (D, L, V) | Alignment axis (x, y, z) | Impact relative to sensor | $A_{MAX}$ (g) | | Peak magnitude (g) |
|---|---|---|---|---|---|---|---|---|---|
| | | | | | | | 10 ms | 30 ms | |
| 1 | 52 | 11.5 | M | L | x | Direct | 102.52 | 64.72 | 213.90 |
| 2 | 52 | 11.5 | T | L | x | Indirect | 98.32 | 51.95 | 145.23 |
| 3 | 52 | 11.5 | M | V | y | Direct | 102.12 | 56.17 | 197.47 |
| 4 | 52 | 6.8 | M | L | x | Direct | 69.59 | 37.69 | 158.73 |
| 5 | 52 | 6.8 | M | L | x | Indirect | 53.48 | 26.13 | 81.32 |
| 6 | 52 | 6.8 | M | V | y | Direct | 65.39 | 35.42 | 175.01 |
| 7 | 76 | 5.0 | M | L | x | Direct | 43.99 | 25.50 | 107.38 |
| 8 | 76 | 5.0 | T | L | x | Indirect | 33.12 | 16.19 | 39.27 |
| 9 | 76 | 5.0 | M | V | y | Direct | 45.98 | 25.20 | 132.60 |
| 10[a] | 76 | 5.0 | M | L | x | Direct | 42.89 | 25.25 | 77.03 |
| 11[a] | 76 | 5.0 | T | L | x | Indirect | 34.76 | 19.68 | 44.07 |
| 12[a] | 76 | 5.0 | M | V | y | Direct | 38.31 | 24.85 | 103.46 |

**Notes.**
[a] The Gelfish model used in trials 10 to 12 was the same as trials 7 to 9 except the surrogate skin was removed from the model prior to strike.

of injury and death among rainbow trout (*Bevelhimer et al., 2019*; *Saylor et al., 2020*), and also based on relative proximity to the embedded accelerometer. We considered impacts a "direct" sensor strike when the blade made contact with the model at the approximate center of the accelerometer. Alternatively, an "indirect" impact was considered any strike where the blade made contact with the model posterior (towards caudal fin) and away from the accelerometer (Fig. 3). All of these conditions were used to assess the ability of the accelerometer to detect differences in strike impact location on our model, which is impossible to determine on live fishes that pass through hydropower turbines. All strikes were recorded at 1000 fps with a high-speed video camera (Model IL4, Fastec Imaging, San Diego, California, USA; fastecimaging.com) and integrated stroboscope LED lighting system (Monarch Nova-Pro 300, Monarch Instrument, Amherst, New Hampshire, USA; monarchinstrument.com) for later review and to confirm blade strike impact velocity.

Data acquisition from the 3-axis accelerometer was initiated immediately prior to engaging the simulated blade strike apparatus. Estimated blade impact velocity with Gelfish was calculated using the running average of the previous 10 frames (*e.g.*, 10 msec) prior to and including impact. Following blade strike, acceleration data were averaged over 10 ms and 30 ms intervals. Maximum acceleration ($a_{MAX}$) was determined using the following equation:

$$a_{MAX} = MAX\left[\frac{1}{t_2 - t_1}\int_{t_1}^{t_2} a(t)\,dt\right] \tag{2}$$

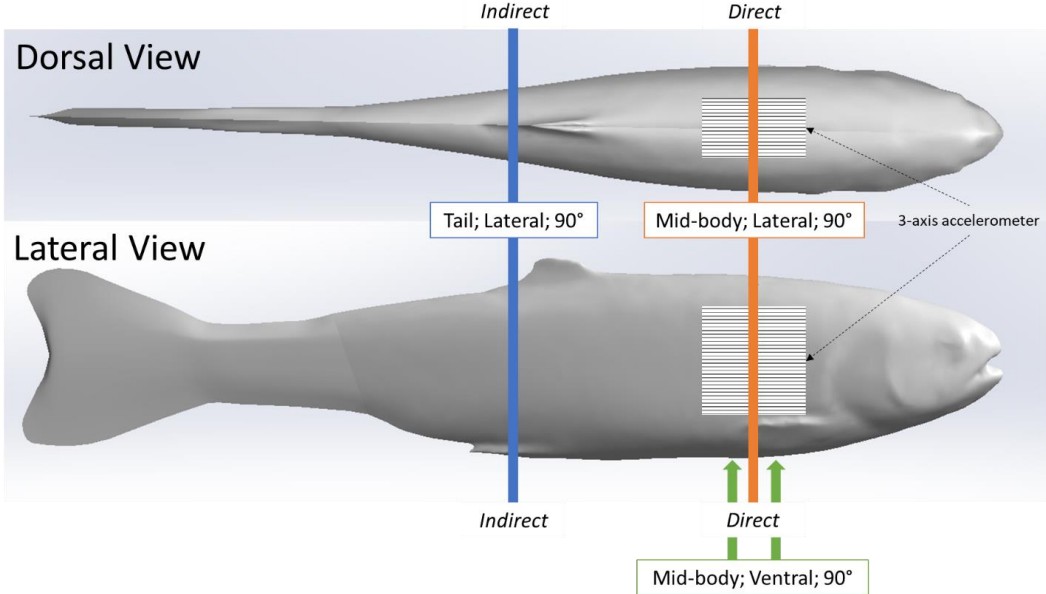

**Figure 3 Location of simulated blade strike impacts on Gelfish model relative to the embedded sensor.** Relative location of 12 blade strike impact trials performed on our rainbow trout Gelfish model. Direct impacts were associated with mid-center strikes to the 3-axis accelerometer embedded posterior to the operculum. Indirect strikes occurred near the caudal fin so that any acceleration was the result of movement following strike. Vertical lines indicate the relative location and orientation combinations we tested including mid-body lateral strikes (orange), tail lateral strikes (blue), and mid-body ventral strikes (green) –all strikes occurred at a 90° angle relative to the longitudinal axis of the model.

according to time $t$ and the desired time interval $t_1$ to $t_2$ during the acceleration pulse, which is reported as acceleration of gravity ($g$). We estimated maximum acceleration using 10 and 30 ms running average intervals to test which interval best captured trends in acceleration. Maximum acceleration is a running average derived from National Highway Traffic Safety Administration (NHTSA) specifications for head injury criteria when using one accelerometer (*Eppinger et al., 1999*). Observed acceleration ($\alpha_{obs}$) was converted to overall magnitude (across all three axes) according to the following equation:

$$\alpha_{obs} = \sqrt{\alpha_x^2 + \alpha_y^2 + \alpha_z^2} \qquad (3)$$

with observed values of gravitation acceleration for the $x$-axis ($\alpha_x$), $y$-axis ($\alpha_y$), and $z$-axis ($\alpha_z$) at each time point, which was plotted as 10 ms and 30 ms running averages of observed acceleration against time (ms). Plots of acceleration were used to determine the relative difference in magnitude between strike impact scenarios (Table 2). In addition, we attempted to link changes in acceleration to rates of injury and mortality reported from previous blade strike impact experiments performed on live rainbow trout (*Saylor et al., 2020*).

## RESULTS

### Ballistic gelatin experiments

The ballistic gelatin concentration could be easily modified to account for different tissue durometers when prepared at 45 °C and a standardized durometer measurement protocol was used. In fact, average durometer °C significantly increased with every 5% increase in ballistic gelatin (prepared at 45 °C) across the entire range tested (one-way ANOVA, $F_{4,10} = 162.40$, $p < 0.001$). We also detected a significant (one-way ANOVA, $F_{1,13} = 532.22$, $p < 0.0001$) relationship between ballistic gelatin concentration and average durometer given by the following linear model:

$$D_{30} = 1.48 \times BG + 0.33 \tag{4}$$

where $D_{30}$ is the durometer following 30 min of warming and $BG$ is the percentage of ballistic gelatin. Ballistic gelatin (prepared at 45 °C) concentration explained ∼97% of the variation in average durometer of the linear model ($R^2 = 0.974$; Fig. 4). Preparation temperatures of 45, 55, and 65 °C did not significantly impact the durometer of our 25% ballistic gelation samples and all three temperatures produced an average durometer of ∼35 units. Warming time significantly (one-way repeated measures ANOVA; $F_{14,28} = 378.96$, $p < 0.001$) impacted average durometer for 25% ballistic gelatin prepared at 45 °C after 10 min of warming at room temperature (22.1 °C) according to Benjamini–Hochberg multiple comparison tests. Average durometer continued to decrease significantly in a linear fashion every 10 min for the first hour of warming except between the 20 to 30-min time period. The average durometer continued to decrease after each warming period but was not significant again until it warmed for 90 min. Eventually, average durometer reached its nadir near 44 units after 150 min of warming where it plateaued for the remainder of the warming experiment (Fig. 5). Ballistic gelatin temperature increased quickly to 19 °C within the first 60 min of warming and did not increase above 20 °C for the remainder of this experiment (Fig. 5).

The use of a surrogate skin increased average durometer of ballistic gelatin blanks (Table 3) and initial Gelfish models (Table 4). Addition of just one layer of surrogate skin significantly (one-way repeated measures ANOVA; $F_{4,8} = 323.96$, $p < 0.001$) increased average durometer by ∼10 units, according to a Benjamini–Hochberg pairwise comparison with samples without surrogate skin. Each additional layer applied to the ballistic gelatin samples also significantly increased durometer, except between two and three layers, which were both near 57 units (Table 3). Up to four layers of surrogate skin caused average durometer to increase by nearly 20 units, to $60.2 \pm 0.9$ units, compared to samples without surrogate skin (average durometer = $42.3 \pm 0.5$). Gelfish without surrogate skin ($36.2 \pm 0.6$) had a significantly (two-tailed, dependent $t$-test; t = -22.209, $p = 0.002$) lower average durometer than the Gelfish model with surrogate skin ($61.6 \pm 1.4$; Table 4). Similarly, bluegill sunfish with scales removed ($54.0 \pm 3.2$) had significantly (two-tailed, dependent $t$-test, $t = 4.9391$, $p = 0.039$) lower average durometer than bluegill with scales intact ($66.8 \pm 1.8$; Table 4). Lastly, average durometer of the Gelfish model with a surrogate skin was statistically indistinguishable from bluegill samples with scales according to a two-tailed, independent $t$-test (Fig. 6).

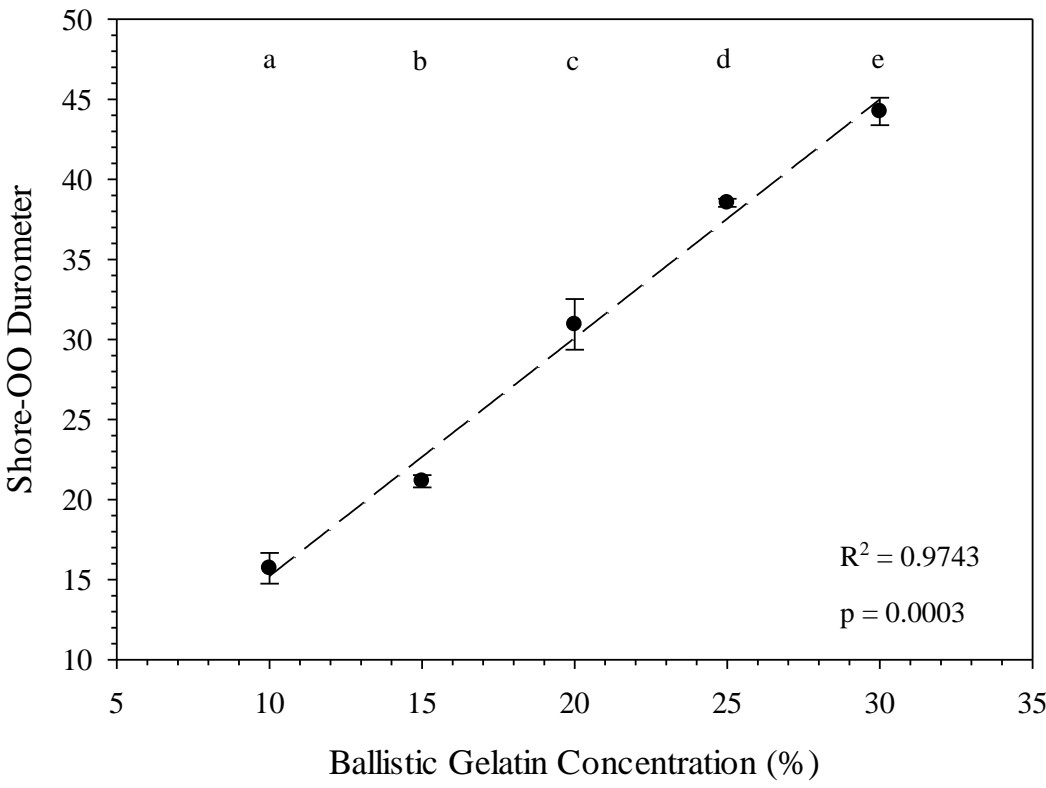

**Figure 4** **Ballistic gelatin concentration (%)** *versus* **average Shore-OO durometer.** Average Shore-OO durometer *versus* ballistic gelatin concentration. The dashed line (——) represents a significant linear regression model ($F_{1,14} = 532.22$, $p < 0.0001$, $r^2 = 0.9743$) fit to these data. Concentration groups with different letters indicate a significant difference according to Benjamini–Hochberg pairwise comparisons which assumed $\alpha = 0.05$.

## 3D scanning and printing fish molds

We successfully laser-scanned and printed molds for four species while a fifth was successfully printed from CT scan data. Scanning frozen fish in an upright position and use of Geomagic Design X software decreased image post-processing time from nearly 40 h (manual processing) down to only 2 to 3 h (with Geomagic software). The resulting SolidWorks models contained more surface features for the rainbow trout *versus* the bluegill, which required markedly more processing time (Fig. 7). The SolidWorks surface models produced using this method were also easier to upload and it was easier to modify features such as fin thickness prior to printing to ensure the resulting ballistic gelatin model did not tear (Fig. 8). The time required to complete 3D printing of each mold varied by species (*i.e.,* smaller species took less time) but was between 8 to 12 h. Printing molds upright (*versus* lying flat) was necessary to limit warping of the mold halves from thermal stresses and ensured the mold halves sealed completely during casing. Acetone sealing successfully prevented ballistic gelatin infiltration through the mold, which reduced cleaning and ensured consistent casting for each model. Additional mounting brackets were included on both the dorsal and ventral surface of the final mold, which allowed

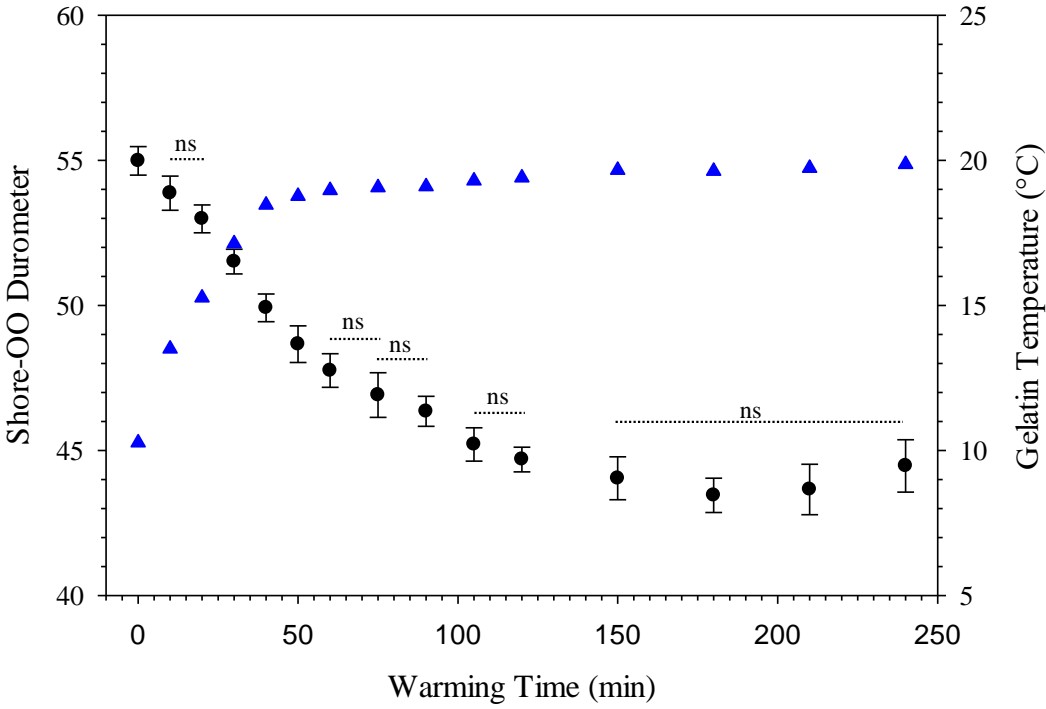

**Figure 5** **Change in Shore-OO durometer and gelatin temperature (°C) caused by periods of warming at room temperature (22.1 °C).** Changes in average Shore-OO durometer (●) and gelatin temperature (▲) as a function of warming time (min). Bars for average durometer represent standard error of the mean. Average durometer decreased significantly (except between time periods indicated with dotted lines which were not significant; ns) as gelatin samples warmed according to one-way repeated measures ANOVA ($F_{14,28} = 378.96$, $p < 0.001$) and Benjamini–Hochberg multiple comparison tests assuming $\alpha = 0.05$. Ambient temperature was 22.1 °C during experimentation.

the accelerometer to be suspended within the ballistic gelatin using a monofilament line (Fig. 9). Multiple Gelfish were cast from the same mold and there is no indication that casting multiple models deteriorated any of the molds. Total preparation time was ∼12 h, including casting (1.5 h), model refrigeration at 4 °C (8 h), and application of four layers of surrogate skin (2.5 h) to the model prior to testing. Many Gelfish models could be created during this time if multiple molds were available.

## Gelfish model testing

The Gelfish model was capable of withstanding multiple blade strike impacts at comparably high velocities (*i.e.,* up to 11.5 m/s) without deteriorating. The rainbow trout model was successfully exposed to nine different impact scenarios before the skin separated from gelatin model; however, the accelerometer remained functional for all 12 strike tests. While the surrogate skin did separate from the gelatin during testing, the gelatin did not deteriorate and could be reused after reapplying skin layers. Similarly, the accelerometer maintained its functionality and could also be cast into another model fish. The flexibility of the model also mimicked actual rainbow trout struck under the same conditions (*i.e.,* mid-body, lateral strikes with 52-mm blade at ∼6.8 m/s); however, overall body curvature

**Table 3** **Results of a one-way repeated measures ANOVA ($F_{4,8} = 323.96$, $p < 0.001$) on average durometer versus number of surrogate skin layers applied to ballistic gelatin samples.** Durometer is presented as average ± standard error (SE) for each skin-layer group ($n = 3$ replicates per group). Skin layer groups with different letters indicate a significant difference according to Benjamini-Hochberg multiple comparison tests. All statistical decisions were based on $\alpha = 0.05$.

| No. of surrogate skin layers | Durometer (± SE) | Significance |
|---|---|---|
| None | $42.3 \pm 0.5$ | a |
| 1-layer | $53.8 \pm 0.4$ | b |
| 2-layers | $57.0 \pm 0.1$ | c |
| 3-layers | $57.3 \pm 0.4$ | c |
| 4-layers | $60.2 \pm 0.4$ | d |

**Table 4** **Results of statistical tests on durometer for Gelfish models and intact bluegill samples.** Durometer is presented as average ± standard error (SE) for each group ($n = 3$ replicates per group). Significance tests included paired (dependent) or unpaired (independent) t-tests. Groups with different letters were considered statistically significant based on $\alpha = 0.05$. Paired t-tests were only performed between Gelfish (skin versus no skin) or bluegill (intact versus without scales) groups, while one unpaired t-test was used to compare average durometer of Gelfish with skin to intact bluegill.

| No. | Group of Interest | Durometer (± SE) | Paired-t | Unpaired-t |
|---|---|---|---|---|
| 1 | Gelfish no skin | $36.2 \pm 0.6$ | a | – |
| 2 | Gelfish with surrogate skin | $61.6 \pm 1.4$ | b | c |
| 3 | Bluegill intact | $66.8 \pm 1.8$ | x | d |
| 4 | Bluegill without scales | $54.0 \pm 3.2$ | y | – |

appeared to be slightly more pronounced with the Gelfish model (Fig. 10). For example, body curvature of Gelfish was noticeably more pronounced during (+0.014s) and after (+0.024s) blade strike impact. The model also followed a similar trajectory out of the holding brackets following blade strike impact, which mimicked trials on live rainbow trout. The surrogate skin also allowed the model to maintain its integrity throughout the impact process, which ensured the entire model (head to tail) reacted similarly to real fish.

Changes in acceleration were detected in all three axes, including just prior to impact, during impact, and as the model moved away following contact with the blade (Fig. 11). Peak magnitude generally occurred 10 ms after the bow wave produced by the blade pushed the model prior to impact. The entire impact sequence took less than 30 ms to complete. The highest peak magnitude and maximum acceleration were detected from a mid-body lateral strike with a 52-mm blade moving at 11.5 m/s (Table 2; Fig. 11). All indirect strikes had noticeably lower peak magnitudes and maximum accelerations, regardless of other strike impact scenarios. Direct impacts to the mid-body ventral surface produced comparable levels of acceleration as mid-body lateral strikes and only differed in the main axis of movement caused by the strike, *i.e., x*-axis *versus z*-axis, respectively. Strikes with the same blade moving slower also had noticeably lower magnitudes—158.73 and 107.38 for the 52-mm blade moving 6.8 m/s and 76-mm blade moving at 5.0 m/s, respectively (Fig. 12). Maximum acceleration detected with a 10 ms time interval was always higher than

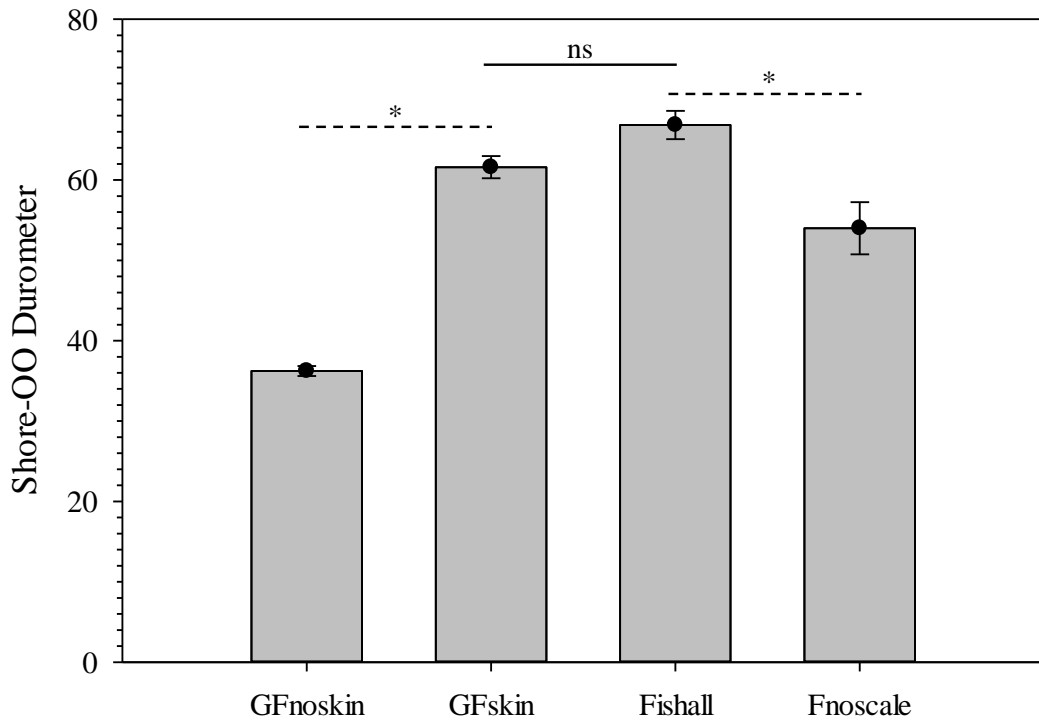

**Figure 6 Bar plots of average Shore-OO durometer for both Gelfish model and bluegill sunfish samples.** Average Shore-OO durometer for one of four groups including Gelfish with no surrogate skin (GFnoskin) or with surrogate skin (Plasti Dip; GFskin) *versus* actual bluegill sunfish, *Lepomis macrochirus*, that were intact (Fishall) or with scales removed (Fnoscale). Average durometer is reported with standard error of the mean for each group ($n = 3$ samples per group). Dashed lines (- - -) represent comparisons between average durometer using two-tailed, dependent $t$-tests while the solid line (——) refers to a two-tailed, independent $t$-test between treatment groups. Note: results of statistical tests were considered significant (*) or not (ns) assuming $\alpha = 0.05$.

acceleration averaged across 30 ms, regardless of group. Gelfish trials completed without surrogate skin (Trials 10 to 12) had lower values than the same trial performed on the Gelfish model with an intact surrogate skin (Trials 7 to 9; Table 2). Trends in magnitude and maximum acceleration suggest that the Gelfish model is also capable of detecting differences in impact scenarios, *i.e.,* indirect strikes *versus* strikes at slower velocities or with thicker blades. A more detailed analysis of correlation with injury risk and mortality was not possible given that only one Gelfish model was tested.

## DISCUSSION

Ballistic gelatin (25% prepared at 45 °C) was used successfully to mimic whole-body tissue firmness of real fish, *i.e.,* bluegill sunfish (Fig. 6). Furthermore, gelatin concentration can be easily modified to account for differences in durometer (15 to 45 units; Fig. 4) among species associated with anatomical disparities in scales and/or muscle tissue. Durometer also appears to be a reproduceable means of estimating the material properties and biofidelity of ballistic gelatin compared to real fish tissue. Durometer varied significantly as a result

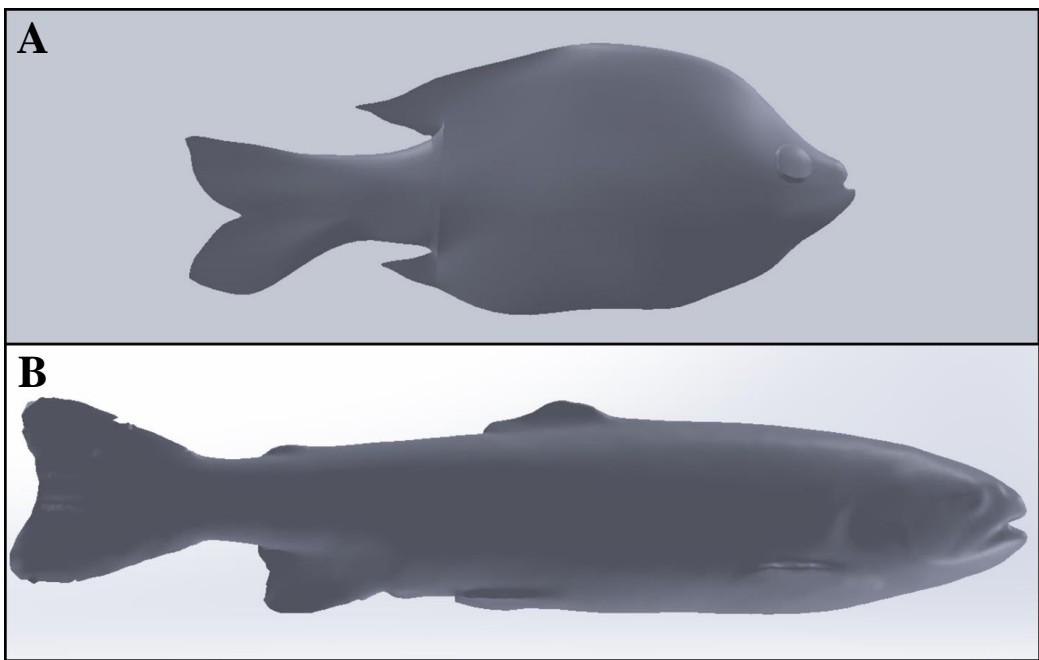

**Figure 7 Comparison of SolidWorks surface models used to produce unique Gelfish models.** Solid-Works surface models of bluegill sunfish (A) created after nearly 40 h of manual user manipulation compared to rainbow trout (B) created in less than 3 h by using Geomagic Design X software to automatically render images and remove unwanted background features from point cloud files. Major landmarks on the bluegill were restricted to the eye, mouth, and the dorsal, caudal, and anal fins. Additional landmarks are visible on the rainbow trout model including eye, mouth, and operculum as well as dorsal, adipose, caudal, anal, pelvic and pectoral fins which are useful for properly embedding each sensor.

of warming, so experimental protocols must be standardized to ensure measured values can be compared, *i.e.,* we used a 30-minute warming time at room temperature. The exact warming time does not matter provided it is used consistently during experimentation; however, warming in excess of 60 min may cause evaporative water loss and shrinkage of the gelatin. No change in average durometer was detected based on preparation temperatures up to 65 °C for 25% ballistic gelatin, but we suggest a 45 °C (or lower) preparation temperature is ideal because additional heating is unnecessary. Temperatures greater than 65 °C may cause detrimental changes to the material properties of ballistic gelatin prepared at lower concentrations of 10 or 20% (*Cronin & Falzon, 2011*; *Maiden et al., 2015*). Use of cinnamon oil increased the usable shelf-life of the Gelfish samples, but refrigeration was still required to avoid evaporative water loss associated with prolonged warming or air exposure. Plasti Dip applied over the ballistic gelatin created models that more closely mimicked the durometer of our whole-fish samples, and the number of layers could be used to further refine durometer as necessary (Table 3). The addition of simulated skin also maintained body shape integrity during blade strikes. Overall, ballistic gelatin appears to mimic tissue properties well, is non-toxic and easy to handle, and produces transparent models that are well-suited for implantation of additional sensors.

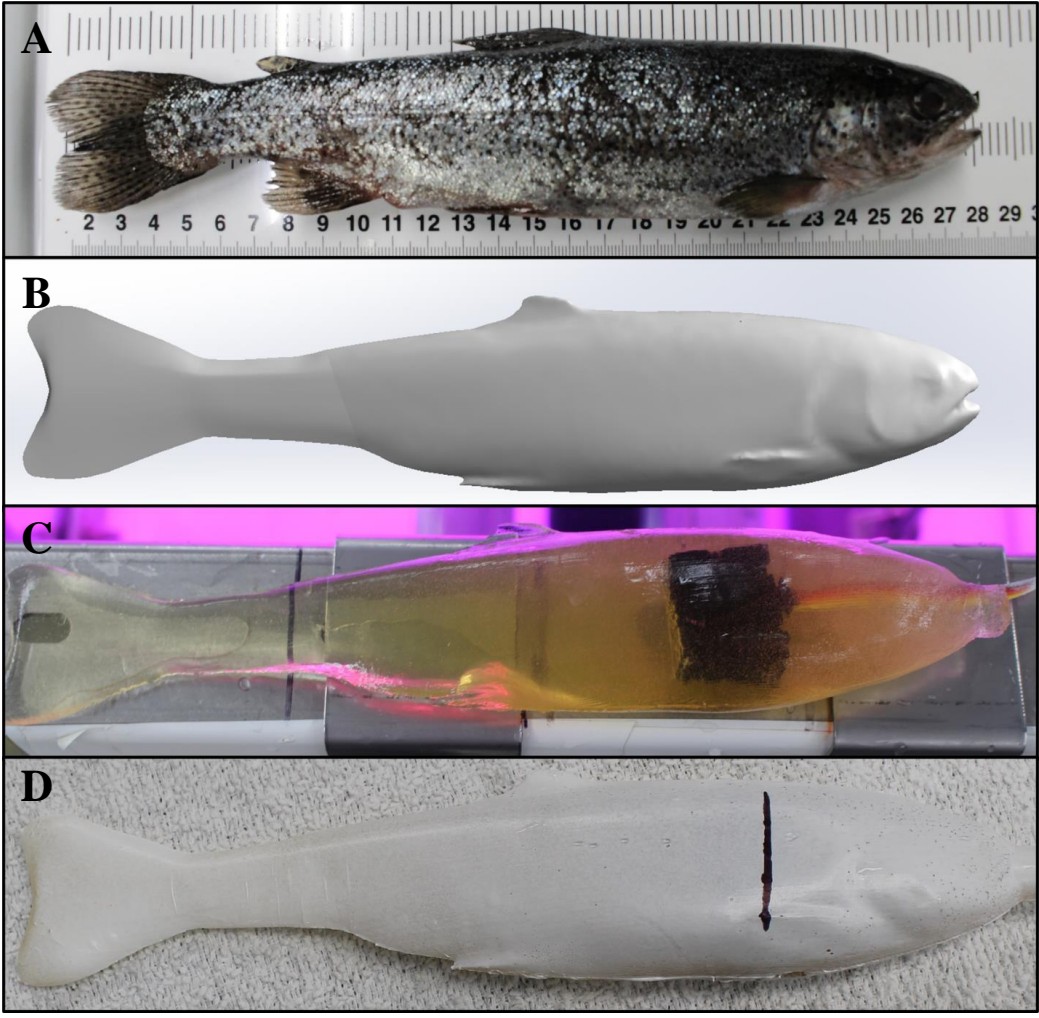

**Figure 8** **Production of a Gelfish model including the scanned fish, SolidWorks surface model, unfinished model without surrogate skin, and completed rainbow trout model with skin.** Series of photographs showing the transition from (A) real rainbow trout, (B) to the final Solidworks surface model (*i. e.,* removing most fins and thickening the caudal fin and peduncle), (C) Gelfish model containing one three-axis accelerometer, and (D) the final Gelfish model with four layers of simulated (*e.g.*, Plasti Dip) skin. Image C also shows the wire connecting the accelerometer to the external data acquisition system. The vertical black line in image D is the approximate center point of the embedded accelerometer used to choose target areas during model testing.

Our scanning techniques successfully created realistic 3D models and molds of multiple species that captured species-specific differences in external morphology. To our knowledge, this is first use of high-fidelity laser and computed tomography scans to design and produce a mold of an entire organism from which to cast a biomimetic model. To date, use of additive manufacturing for creation of biofidelic models has mostly centered around 3D printing the desired animal model directly from scanned data (*Rhyne et al., 2017*; *Walker et al., 2019*; *Tetzla et al., 2020*). The cost of printing multiple fish models directly is far greater than multiple models cast from just one mold; consequently, the

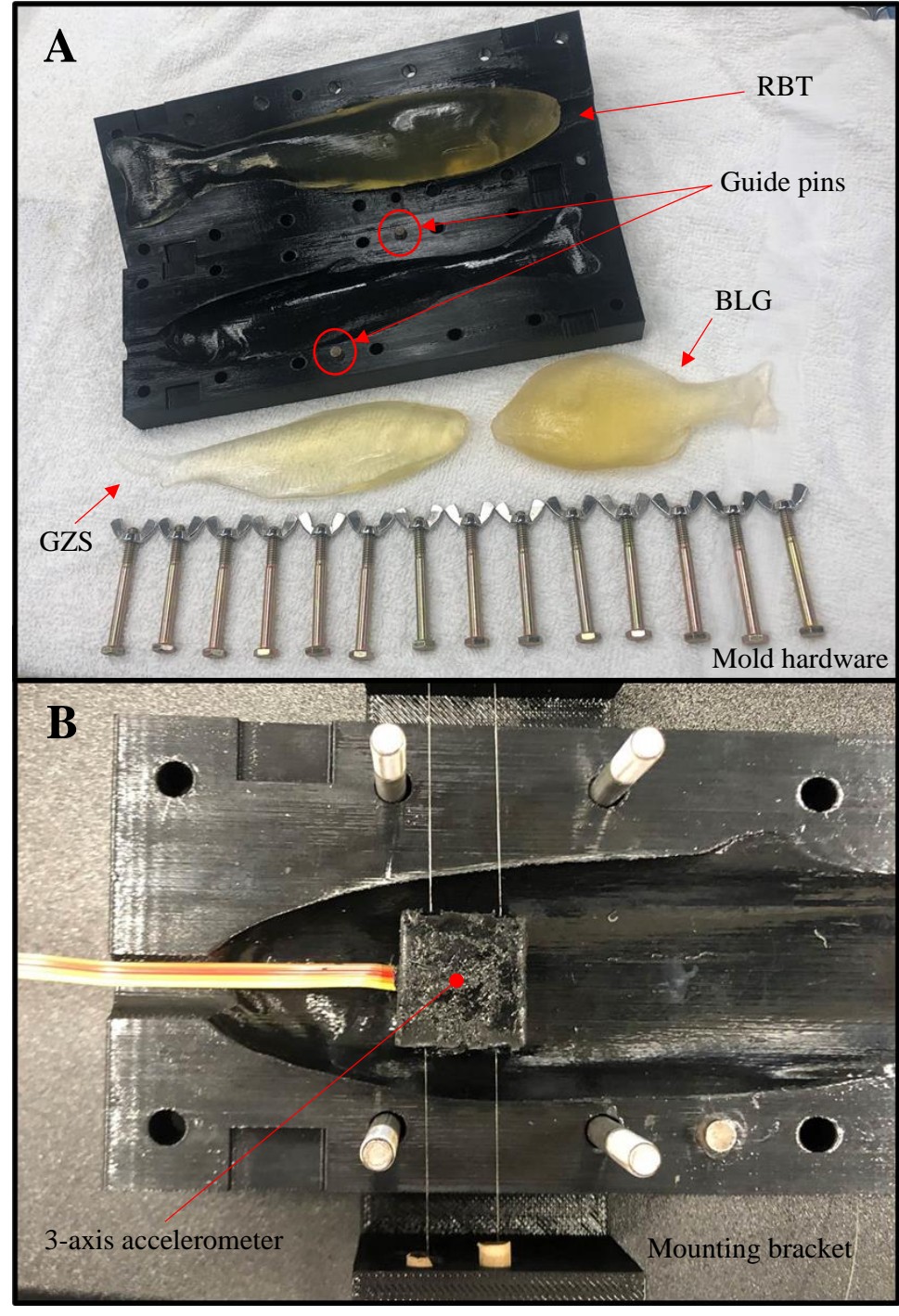

**Figure 9** **Gelfish mold and associated hardware used to seal the mold halves and embed a sensor.** Image (A) represents the completed rainbow trout mold including hardware and guide pins (red circles) used to completely close and seal each half of the mold. Image (B) shows the anterior region of the rainbow trout mold including a mounting bracket (also 3D printed) used to guide monofilament tethers that held the potted accelerometer in place during casting. Note: completed Gelfish models (ballistic gelatin only) of rainbow trout (RBT), gizzard shad (GZS), and bluegill (BLG) are also shown in image A.

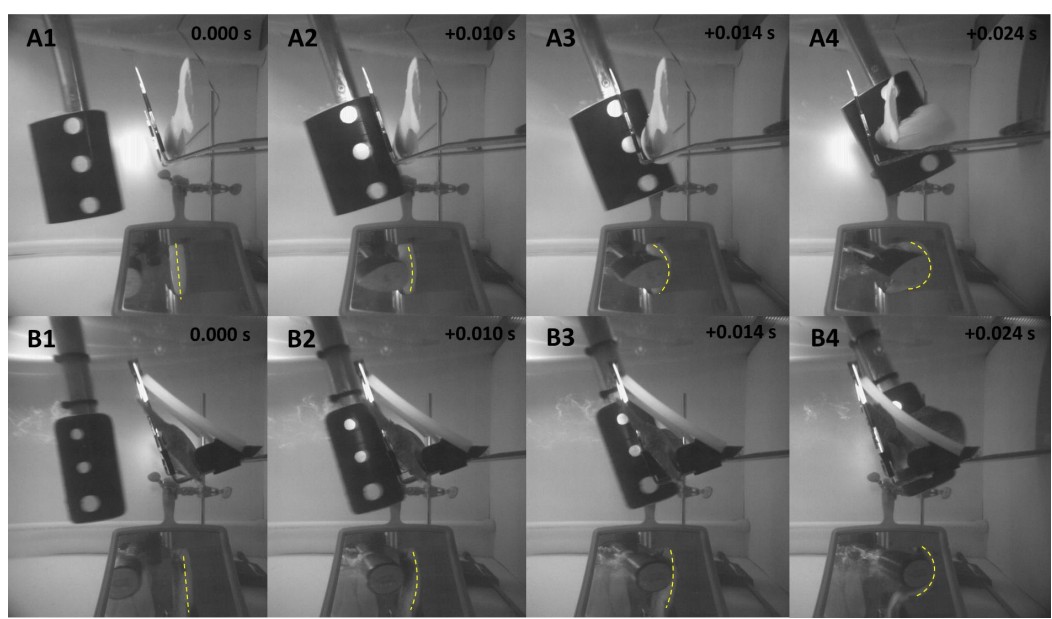

**Figure 10 Highspeed images before, immediately prior to, during, and after contact with the simulated hydropower turbine blade.** Highspeed video (1000 fps) images of the A) Gelfish model and B) sub-adult rainbow trout being struck with a 52-mm blade at 6. 8 m/s at approximately the same location on the mid-body lateral surface at 90°. Dashed lines were included to show body curvature of the ventral surface of both the model and trout in each frame including blade approach (0.000 s; reference), just before contact (+ 0.010 s), at impact (+ 0.014 s), and through maximum curvature following impact (+ 0.024 s).

additive manufacturing industry has focused on printing molds which are less labor intensive, cheaper to produce, and of comparable durability to traditional sand-cast molds (*Hassen et al., 2016*; *Hassen et al., 2020*; *Hawaldar & Zhang, 2018*). Freezing fish worked well for scanning purposes to minimize movement of the specimen during scanning. The FARO scanning system was the easiest to use and produced a high-fidelity rendered model in about 10 min. In contrast, the Leica laser tracker and scanning system was very sensitive to slight deviations in fish positioning (caused by thawing) which made image rendering more difficult and increased post-processing time. Mounting the frozen fish on a turntable and securing the laser scanner may help decrease scanning time without compromising the quality of the 3D rendered images. Use of the software Geomagic Design X decreased post-processing time and produced a final CAD model with more realistic landmarks (Fig. 7B) compared to a model that required 40 h of manual image processing (Fig. 7A). Computed tomography scans of a small American eel (scanned at UTCVM) were also used to create a small and large eel mold. Similar CT scan data from online repositories were not always useful because many images only captured skeletal features and excluded soft tissues (*e.g.*, muscle and skin) which are necessary to model body shape. The success of our 3D scanning and printing techniques suggests these methods can accurately recreate the desired features of any organism scanned directly or rendered from scans available *via* online databases.

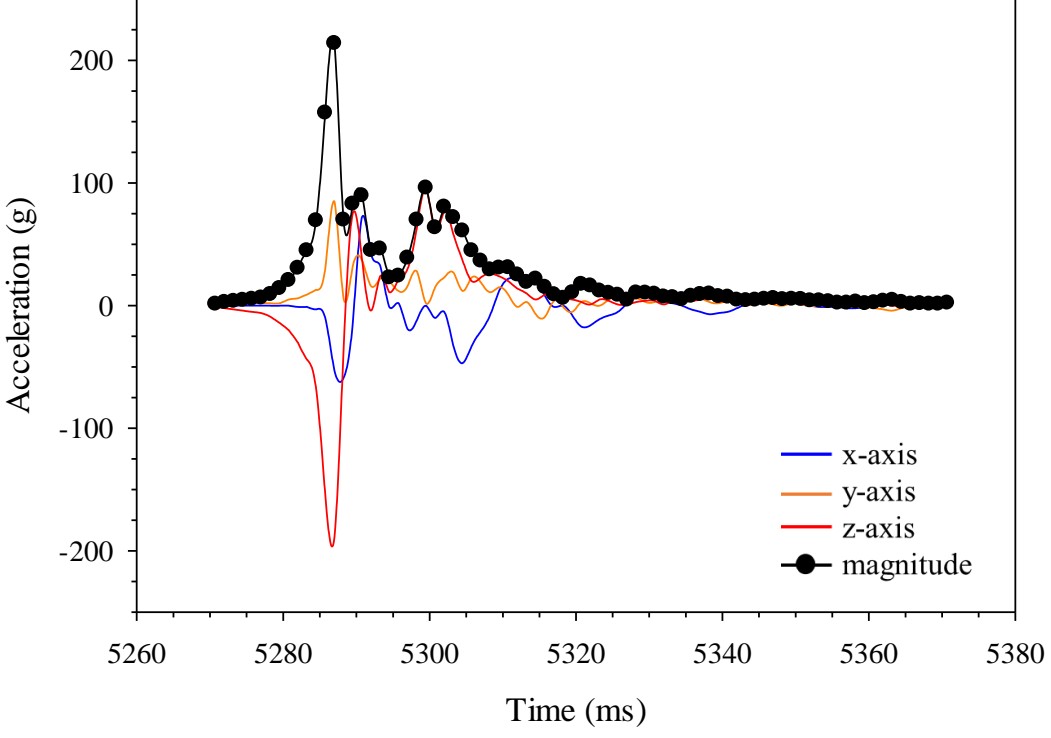

**Figure 11** **Changes in gravitational acceleration in three dimensions during simulated blade strike testing of the Gelfish model.** Example plot of acceleration (g) for the Gelfish model struck with the 52-mm blade on the mid-body, lateral surface at 11. 5 m/s. Magnitude was calculated across all three axes for each time step and reached a peak of nearly 220 g in this trial (#1; Table 2).

Gelfish responses were similar to real fish with respect to overall flexibility during simulated impact trials. The model began to bend immediately prior to impact, followed by whole-body curvature during impact, and free movement after the impact sequence (Fig. 10) which is similar to responses observed in rainbow trout laboratory trials (*EPRI, 2008*; *EPRI, 2011*; *Bevelhimer et al., 2019*; *Saylor et al., 2020*). Body curvature was observed as a spike in acceleration in the *z*-axis (*e.g.*, lateral, side-to-side movement) and tumbling of the model after impact was observed as noticeable changes in acceleration across all three axes (Fig. 11). The surrogate skin (Plasti Dip) enhanced overall Gelfish performance by adding stiffness to the model. Analysis of high-speed videos suggested that the amount of curvature in the Gelfish model may have exceeded that of real rainbow trout in its current form (Figs. 10A4 and 10B4). Additional flexibility in our model is likely because it lacks an endoskeleton, overlapping myomeres, and imbricated scales of an actual fish which impose limits on natural flexibility. The number and size of vertebral centra, specifically, has a profound effect on flexibility (or stiffness) among fish (*Lindsey, 1978*; *Long & Nipper, 1996*; *Brainerd & Patek, 1998*) and inclusion of a simulated vertebral column could better mimic natural flexibility. In addition, the lack of a vertebral column and/or other support elements caused a delayed response in the movement of the tail compared to the body of the model, following contact with the blade. The Gelfish model was successfully struck 12 times

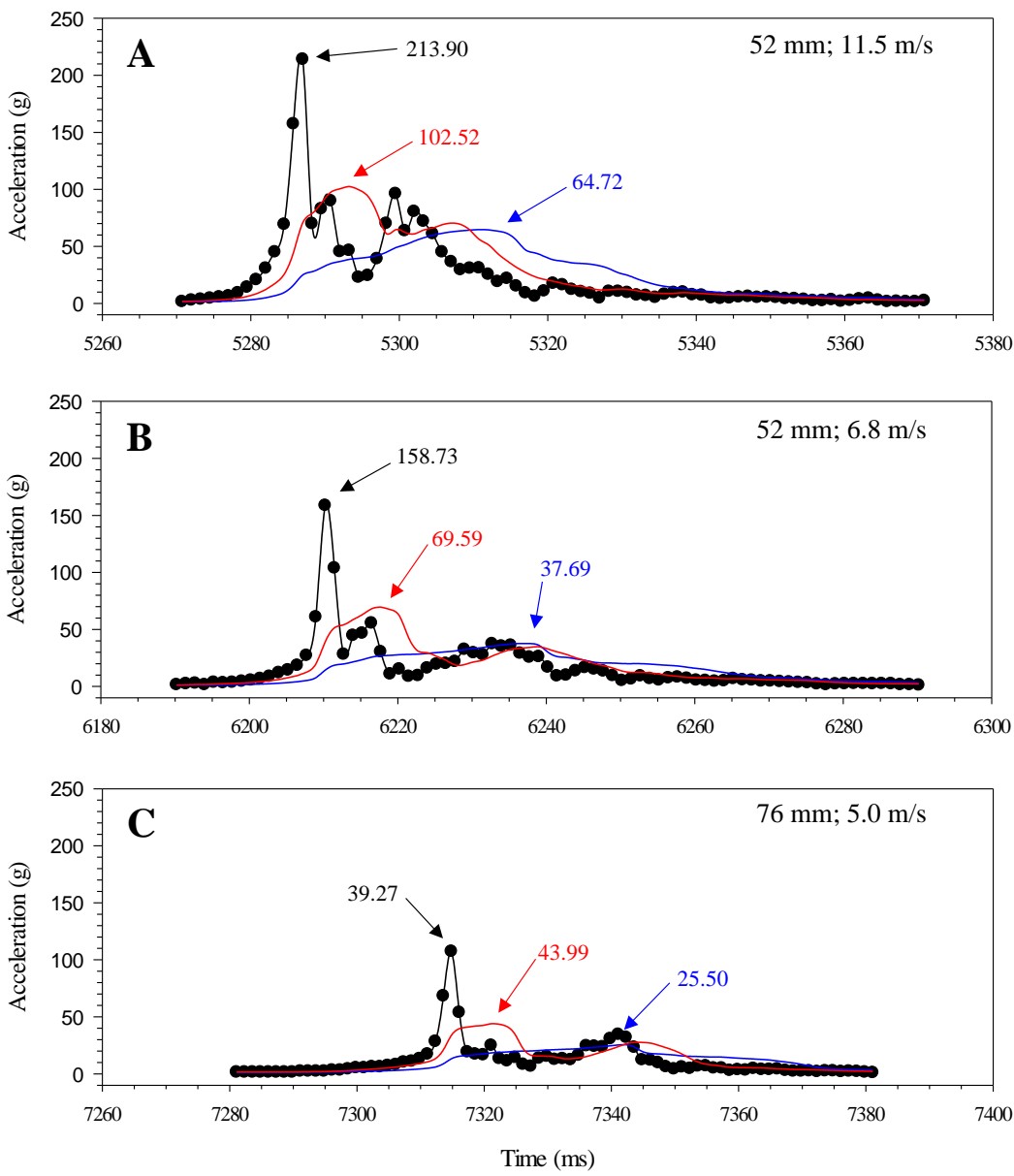

**Figure 12 Changes in gravitational acceleration associated with simulated blade strike testing at the same location on the Gelfish model with blade leading edge widths and impact velocities.** Three plots of overall magnitude (black lines) and 10 ms (red) or 30 ms (blue) running average of acceleration *versus* time (ms). Results from three impact scenarios are shown including mid-body lateral strikes at 90° and (A) the 52-mm blade moving at 11.5 m/s, (B) 52-mm blade moving at 6.8 m/s, or (C) 76-mm blade moving at 5.0 m/s. Numbers reported with each curve include peak magnitude or maximum acceleration.

without disintegrating, but the surrogate skin eventually separated from the gelatin and was removed after the ninth strike trial. The latter suggests that our model could be used more than once without losing its structural integrity and while maintaining consistent responses to multiple impact scenarios. More detailed insights into model behavior or flexibility are

not warranted because only one model was tested; however, the overall performance and response of Gelfish compared to actual fish supports the biofidelity of this prototype model.

The single 3-axis accelerometer worked well to capture changes in acceleration that a fish may experience during impacts from turbine blades. We detected changes in acceleration during all aspects of the blade impact sequence, including a rise in acceleration as the blade approached, a peak during impact with the model, and random changes in all axes as the model tumbled after impact (Fig. 11). The greatest changes in acceleration co-occur with lateral bending of the model along the $z$-axis, observed during review of high-speed videography (Fig. 10). While we only tested one complete model, there were notable changes in absolute magnitude and time-averaged acceleration associated with blade leading-edge width, impact velocity, and orientation of the model (Table 2). More specifically, faster velocities and the thinnest blade had the highest observed changes in peak and time-averaged acceleration—these conditions are also thought to be the most injurious and lethal to rainbow trout struck by turbine blades (*EPRI, 2008*; *EPRI, 2011*; *Bevelhimer et al., 2019*; *Saylor et al., 2020*). Trends in time-averaged acceleration (both 10 and 30 ms) also detected higher changes in peak magnitude as a result of mid-body lateral strikes, compared to both tail lateral and mid-body ventral strikes, which is consistent with estimated mortality rates for this species (*Bevelhimer et al., 2019*; *Saylor et al., 2020*). The exact relationship between acceleration and probability of injury or mortality has yet to be determined; however, development of injury criteria and/or probability of fracture models, similar to automobile safety tests (*Faerber & Kramer, 1985*; *Digges, 1998*; *Eppinger et al., 1999*; *McHenry, 2004*), may help connect accelerometer data to laboratory dose–response relationships. Our sensor detected similar estimates of peak acceleration as the Sensor Fish package (*i.e.,* 213 and 223 g, respectively) struck under the same conditions, and at a higher velocity of 7.5 m/s (*Bevelhimer et al., 2019*). The latter suggests our sampling rate of 800 Hz was capable of detecting comparable levels of peak acceleration, given that the Sensor Fish sampling frequency is 2.5 times higher (*Deng et al., 2007b*; *Deng et al., 2014*). More impact trials are needed on multiple Gelfish models to establish repeatability and estimate the variation in peak magnitude before making more detailed comparisons between Gelfish and Sensor Fish.

## CONCLUSIONS

Use of ballistic gelatin and 3D scanning to produce reusable molds worked well to recreate the overall shape and basic biomechanical properties of a real fish. Ballistic gelatin was easy to work with and could be modified to account for small changes in tissue firmness related to different species. Ballistic gelatin does have a limited shelf life (even with preservatives) and the need for refrigeration was important to minimize evaporative water loss. The Plasti Dip surrogate skin also appeared to bond well with ballistic gelatin and its inclusion better captured the natural flexibility of a real fish following impact from a simulated turbine blade. Laser and CT scan image data were successfully used to capture the overall shape and identifying surface details of each fish species. Scanning frozen fish was necessary to limit unwanted movement of the fish, which would significantly increase post-processing

time. We successfully used these scanned images to create and print molds using additive manufacturing, which enabled casting of multiple models with no indication of mold deterioration.

The response of the Gelfish model from simulated impact conditions suggests it may be slightly more flexible than real fish, but more tests are required to quantitively confirm its biomechanical properties. Results of blade strike impact tests suggest that the embedded accelerometer detected changes in acceleration associated with impacts at different velocities, leading edge widths, and locations along the body. These changes were consistent with the responses of actual fish exposed to the same scenarios, *i.e.,* differential rates of injury and mortality as strike conditions change. Changes in time-averaged and peak acceleration will likely be most useful if linked to novel injury criteria or mortality thresholds like those used during impact safety tests in the automobile industry. Initial production of a prototype Gelfish was successful, but more development is necessary to assess its biomechanical accuracy and determine how sensor output may be linked to rates of injury or mortality detected during dose–response testing.

The basic Gelfish model and the process used to create it needs further development to augment its biofidelity and make it more versatile for use in other applications. At the least, additional impact trials on multiple models are needed to establish variation in model responses and assess the replicability of sensor output. While our method can produce any desired species, the same model would be more useful if it accurately represented groups of similar fishes (taxonomically or functionally) defined by the intended application. For example, surrogate species are used to represent taxonomic groups of fishes for blade strike trials, yet the functional or biomechanical relevance of these groups has yet to be addressed (*Saylor et al., 2020*).

Ballistic gelatin worked well to mimic fish tissue but refrigeration was necessary to prevent water loss and warming time affected firmness of the model. Newer versions of this model may seek to create a model using synthetic polymers which can be modified to enhance model biofidelity without the need for refrigeration or preservatives. Further development of a simulated skin, and dedicated inclusion of structures that mimic the materials properties of bone and organs, may also better approximate the natural flexibility and responses of the fish body. All new model developments should be replicated and the biomechanics of impact observed in the model should be compared to that of real fish to maximize biofidelity. Inclusion of more than one accelerometer or the use of new sensors, including strain or fracture gauges, would provide additional information to better link sensor output with biological response data. The next model should also prioritize smaller sensors with higher sampling rates that increase the precision of sensor output while minimizing unnecessary gains in mass to the model. Finally, newer versions of Gelfish would benefit from onboard storage and/or wireless communication technologies to allow it to move more freely and make it recoverable during field tests.

While we developed this model with hydropower applications in mind, our techniques described here may have other applications well. Similar applications might include (1) testing blade strikes associated with irrigation and water pumping stations, (2) strikes from marine hydrokinetic turbines, (3) impacts from boat impellors on large fishes (*e.g.,*

sturgeon or paddlefish) and coastal marine mammals (manatees and whales), (4) mortality of birds and bats caused by impacts from wind turbine blades, and (5) mortality among fish, sea turtles, and other marine life caused by unintended interactions with commercial fishing gear. Regardless, biofidelity remains paramount for future Gelfish development and application, which further distinguishes it from lower-biofidelity technologies, like Sensor Fish, currently used in a similar application.

## ACKNOWLEDGEMENTS

We must first and foremost thank Mark Peterson, Brennan Smith, Eric Pierce, and Stan Wullschleger (Energy and Environment Sciences Directorate Energy Efficiency and Renewable Energy Program, Oak Ridge National Laboratory) for their fervent and continued support of this project from the beginning. We also thank summer intern Clara Layzer for her work and insights during initial development of the Gelfish project. Much thanks are given to the members of the LDRD Seed Money Fund Committee and the anonymous reviewers of our proposal for their feedback and support. Lastly, we thank Teresa Mathews (Group Leader, Biodiversity and Ecosystem Health Group, ORNL) for her technical review and comments on this article.

### Funding

This work was supported by the Laboratory Directed Research and Development (LDRD) Seed Money Fund at Oak Ridge National Laboratory (No. 32102656), managed by UT-Battelle, LLC for the US Department of Energy under contract DE-AC05-00OR22725. The funders had no role in study design, data collection and analysis, decision to publish, or preparation of the manuscript.

### Grant Disclosures

The following grant information was disclosed by the authors:
Laboratory Directed Research and Development (LDRD) Seed Money Fund at Oak Ridge National Laboratory:  32102656.
US Department of Energy: DE-AC05-00OR22725.

### Competing Interests

The authors declare there are no competing interests.

### Author Contributions

- Ryan Saylor conceived and designed the experiments, performed the experiments, analyzed the data, prepared figures and/or tables, authored or reviewed drafts of the paper, and approved the final draft.
- Peter L. Wang conceived and designed the experiments, analyzed the data, performed the computation work, prepared figures and/or tables, authored or reviewed drafts of the paper, and approved the final draft.

- Mark Bevelhimer conceived and designed the experiments, authored or reviewed drafts of the paper, and approved the final draft.
- Peter Lloyd, Celeste Atkins and Brian Post conceived and designed the experiments, performed the computation work, authored or reviewed drafts of the paper, and approved the final draft.
- Jesse Goodwin performed the experiments, performed the computation work, prepared figures and/or tables, authored or reviewed drafts of the paper, and approved the final draft.
- Robert Laughter and David Young performed the experiments, analyzed the data, performed the computation work, authored or reviewed drafts of the paper, and approved the final draft.
- Dustin Sterling performed the experiments, authored or reviewed drafts of the paper, and approved the final draft.
- Paritosh Mhatre performed the experiments, analyzed the data, performed the computation work, prepared figures and/or tables, authored or reviewed drafts of the paper, and approved the final draft.

## Data Availability

The raw data is available in the Supplementary File.

## Supplemental Information

Supplemental information for this article can be found online at http://dx.doi.org/10.7717/peerj-matsci.16#supplemental-information.

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
