# Peer review of "Creation of a prototype biomimetic fish to better understand impact trauma caused by hydropower turbine blade strikes"

_PeerJ Materials Science, doi:10.7717/peerj-matsci.16_

## Round 0.1 · original submission · Major Revisions

Overall, this manuscript describes new and pertinent data for creating a biomimetic fish to replace live fish and collect essential data during hydropower turbine testing. The experimental design and studies provide valuable data to further the progress towards a model with high biofidelity. The experimental design and results presented in the manuscript suggest further progress toward a viable model than is represented. Most of the data support a strong methods article that helps the field continue developing a model with high fidelity. If the authors wish to communicate a functioning model, additional replicates and experimentation is warranted. I believe the authors need to decide whether they should present a methods paper or a fully-develop research article.

Please refer to the reviewer's comments on how to improve and modify the manuscript. Specifically, reviewer one comments #1 and #4. Along these same lines, the authors should replicate the Gelfish testing, as only one fish was tested, as the authors mentioned. Additional fish testing would provide the community more information about how different fish can be modeled (i.e., larger fish like salmon, or smaller such as the bluegill). The authors should consider how vital direct flexibility measurements of the models would increase the study's viability over the indirect methods the authors used to describe the similarity between the model and the real fish.

Additional comments from reviewer 2 to focus on when making revisions include:
1.) Please add more information about the durometer measurements—including calibration and why this particular method provides enough data to avoid other forms of analysis.
2.) Additional comparisons between live fish and the model are warranted to fully understand the testing conditions, the model response, and valid comparisons. The conclusions reached using previously reported data needs strengthening with more testing.
3.) Please elaborate on the decision not to include internal support such as bone and how this might affect the model's biofidelity.

Additional comments to consider:
4.) Please support your decision of fish choice to model with data or references that provide common fish species that interact with turbines.
5.) How do cooling and warming change the hardness of the ballistic gel? Does a freshly prepared model have the same hardness as one that has been stored in a refrigerator, then warmed back to room temperature?
6.) Please comment on the decision for using the model fish after warming for 30 minutes, even though the plot of hardness versus time in Figure 6 suggest that the temperature of the gel is not stable until >50 minutes, and the hardness isn't stable until > 110 minutes.

Reviewer 1 ·

Basic reporting

The manuscript was written in professional English. The research backgrounds was fully described by using the appropriate references.

Experimental design

The contents of the research is in the field of the journal, and the research significance is explained in detail. And the study meets ethical standards.

Validity of the findings

No comment.

Additional comments

The research in this paper is very meaningful. It has potential application value to imitate fish similar to live fish by using bionics. However, there are still some shortcomings in the research or questions that readers may have. As shown below:
(1) How to evaluate the flexibility of Gelfish, is it reasonable to evaluate only by hardness? In this paper, only the hardness can be used to determine whether the tissue similarity between Gelfish and live fish is correct? I think this is not rigorous. I don't know whether other scholars in previous studies also used hardness to characterize it.
The hardness value of the ballistic gelatin in the article should be given, is it Shore A? Or something else?
(2) In the process of preparing the sample, the dosage value of de-foaming agent added should be clear, rather than using a few drops to express it.
(3) Whether it is necessary to vacuum defoam the sample during the fabrication process. I don't know how you solved the air bubbles in the prepared sample. In our actual operation process, vacuum defoaming is required, otherwise there will be bubbles.
(4) Why did you choose bluegill, rainbow trout, gizzard shad, and white bass as the prototype for your study.
(5) The marker points in Figure 1 are similar to those added in the post-image processing process. Please provide the actual biological pictures with marker points during the scanning process.
(6) The scanning accuracy of the scanner in this paper should be given, rather than let the reader find it by himself.
(7) Whether the sensor designed by the author has been calibrated. This is very critical to the accuracy of the sensor.
(8) The authors should check whether the format of literatures meet the requirements of the journal, and check the references in the paper, such as P26 Line 677 and P27 Line 706 the corresponding literatures.
Moreover, the paragraphs of the manuscript should be double-spaced, and the font should be times new roman. The format of the formula should be centered and right aligned.

Reviewer 2 ·

Basic reporting

Clear writing is used throughout, with sufficient literature references. The introduction is focused on biomimicry and modeling as the foundation for this work, though some additional focus on how similar models have been used to make decisions or inform scientific questions may be useful. There is a considerable amount of biomechanics literature on things like using 3D printing to understand form or function of organism teeth, armor, etc, to model how adaptations might be used in the actual organism. Adding this background might expand the applicability of the research.

Article is well structured

This paper is more of a methods paper than one with a self contained hypothesis and results, but the paper has clear objectives and relevant results.

Experimental design

Research question is relevant and meaningful. The topic, understanding how to replicate fish structure and materials, is a knowledge gap. The methods for creating the materials and iterations are well described, with detailed description of how each step was completed.

Additional detail is needed for the durometer measurements. Describe how the model and fish were fixed while taking the measurements. Describe how deep the durometer set up is designed to penetrate. While the durometer is a decent way of understanding the material properties, it might be worth while describing why this one was chosen, as opposed to others, like comparing rated elastic modulus, bending or torsion of the full structure, etc.

Validity of the findings

Findings are generally robust. The results of the strike analysis on the Gelfish is somewhat speculative, as it is not compared to a fish response in the paper, though it is modeled off of previous results. If it were compared to a fish, how does the model assist in understanding whether or not the strike is a lethal strike? What characteristics of the strike are the authors looking for? A certain amount of force or deflection? This is discussed a bit in the conclusions, but it is difficult to understand if biofidelity can be achieved without injuring more actual fish.

While the strike analysis adds value for the potential hydropower industry, I don’t know if the strike analysis adds important conclusions to the results of whether or not the Gelfish has high fidelity as a model. I would recommend strengthening comparisons to the rainbow trout lab trials mentioned in line 562. Throughout this section, compare findings to the expectations of an actual fish.

Line 575 – the paper describes how the Gelfish began to disintegrate after being struck repeatedly. I would guess that a fish would also begin to disintegrate after repeated trauma. A high fidelity model would need to respond to trauma like the fish. In this case, the fact that the skin separated is an important difference, as that would be an unlikely result in the actual fish. I think the paper would benefit from additional focus on describing how the results compare to actual fish in these sections.

Additional comments

Overall, I think this a relevant study with useful data toward creating model fish or other marine animals.

I would like some additional discussion about the lack of bone structure in the model fish. How does this compare to the way an actual fish behaves? What is the justification for not including some internal structure?

Errata:
Line 200: I think plastic storage bag would be better

---

## Round 0.2 · accepted · Accept

We appreciate you taking the time to make the reviewers' and my suggested changes and I am happy to accept the article for publication. We extend apologies to you and your colleagues for the length of time to publication and appreciate your patience during this challenging year.